# GWAS in the southern African context

Yolandi Swart[1], Gerald van Eeden[1], Caitlin Uren[1,2], Gian van der Spuy[1,3], Gerard Tromp[1,2,3], Marlo Möller[1,2]*

**1** DSI-NRF Centre of Excellence for Biomedical Tuberculosis Research, South African Medical Research Council Centre for Tuberculosis Research, Division of Molecular Biology and Human Genetics, Faculty of Medicine and Health Sciences, Stellenbosch University, Cape Town, South Africa, **2** Centre for Bioinformatics and Computational Biology, Stellenbosch University, Stellenbosch, South Africa, **3** SAMRC-SHIP South African Tuberculosis Bioinformatics Initiative (SATBBI), Center for Bioinformatics and Computational Biology, Cape Town, South Africa

☯ These authors contributed equally to this work.
* marlom@sun.ac.za

**Data Availability Statement:** All genetic data used in this paper were simulated or retrieved from public repositories (http://www.internationalgenome.org). All scripts used to generate and conduct data analysis are deposited

## Abstract

Researchers would generally adjust for the possible confounding effect of population structure by considering global ancestry proportions or top principle components. Alternatively, researchers would conduct admixture mapping to increase the power to detect variants with an ancestry effect. This is sufficient in simple admixture scenarios, however, populations from southern Africa can be complex multi-way admixed populations. Duan *et al*. (2018) first described local ancestry adjusted allelic (LAAA) analysis as a robust method for discovering association signals, while producing minimal false positive hits. Their simulation study, however, was limited to a two-way admixed population. Realizing that their findings might not translate to other admixture scenarios, we simulated a three- and five-way admixed population to compare the LAAA model to other models commonly used in genome-wide association studies (GWAS). We found that, given our admixture scenarios, the LAAA model identifies the most causal variants in most of the phenotypes we tested across both the three-way and five-way admixed populations. The LAAA model also produced a high number of false positive hits which was potentially caused by the ancestry effect size that we assumed. Considering the extent to which the various models tested differed in their results and considering that the source of a given association is unknown, we recommend that researchers use multiple GWAS models when analysing populations with complex ancestry.

## Introduction

Differential assortment of ancestral allele frequencies across a genome results from selection, mutation or genetic drift, when previously isolated populations interbreed [1, 2]. Taking advantage of these differences in allele frequencies can help identify population-specific disease risk alleles associated with disease phenotypes due to various ancestries being exposed to distinct environments and pathogens [3]. Admixed populations present unique opportunities to identify ancestry-specific disease risk alleles for various populations simultaneously. African

and publicly available on Github (https://github.com/YolandiSwart/GWAS_SA) and Zenodo (10.5281/zenodo.6984444).

**Funding:** This research was funded (partially or fully) by the South African government through the South African Medical Research Council and the National Research Foundation. GvE was supported by the DST-NRF Innovation Doctoral Scholarship. YS was supported by a Stellenbosch University Postgraduate Bursary. There was no additional external funding received for this study. The funders had no role in study design, data collection, and analysis, decision to publish, or preparation of the manuscript.

**Competing interests:** The authors have declared that no competing interests exist.

ancestries harbour the most genetic diversity and contain more complex linkage disequilibrium (LD) blocks than other continental populations and the precise evolutionary events that shaped their genomes are mostly unknown [4, 5]. When the historical events are unknown, it is often difficult to know beforehand which effect (allele, ancestry or the interaction between them) has the most significant effect on the disease phenotype under study. Therefore, studying admixed African populations presents a rich opportunity for the discovery of ancestry-specific disease risk alleles, however, the complex genomic architecture of admixed African genomes warrants careful consideration of both global and local ancestry.

A genome-wide association study (GWAS) is commonly used to discover associations between single nucleotide polymorphisms (SNPs) and a diverse spectrum of complex traits of interest [6]. Attributing associations at the SNP level, however, requires large sample sizes when the SNP being tested has a small effect on the trait of interest. GWAS is further complicated when individuals have admixture-induced linkage disequilibrium (admixture LD) blocks. Admixture mapping uses the admixture LD blocks inherited from a specific ancestral population to test for an association with the trait of interest [2]. Hence, the ancestry rather than the genotypes are traced in the association process [7] and the SNPs that affect the trait of interest can only be localised to their respective ancestral blocks (presented as an admixture peak on a Manhattan plot). There are also smaller LD blocks within admixture LD blocks that originated prior to admixture that are inherited from the parent populations that contributed to the admixture (ancestry LD). Admixture LD and ancestry LD cause population stratification in a GWAS which reduces the power to detect a significant association and could lead to increased false positive hits and false negative hits [8]. Therefore, at a given locus within an admixed population there will be groups of individuals with similar LD. If one or more of these groups is disproportionately represented within a given phenotype, a false association will exist between that group and that phenotype. Many approaches address this problem by including global ancestry and local ancestry in GWAS models [9–11].

Duan et al. [8] expanded on these approaches by additionally modelling the interaction between allelic effects and ancestry effects. They proposed a two-step approach, called the Local Ancestry Adjusted Allelic (LAAA) testing procedure, that first detects associations by jointly modelling allelic effects, ancestry effects and interaction effects and then determines the source of the association. Their approach proved robust and powerful, but it was only assessed in a two-way admixed population. Therefore, it remains to be determined if this model will be able to capture the correct association signals with minimal power loss and spurious associations (including both false positive hits and false negative hits) in complex multi-way admixed populations exhibiting genetic heterogeneity (i.e. ancestry patterns that differ between admixed populations formed by the same source populations). We believe that testing this will go a long way to identifying robust models that can be used in a wide array of demographic scenarios.

We used the complex demographic history of the Nama and the South African Coloured (SAC) populations to simulate a three-way admixed population (as represented by the Nama) and a five-way admixed population (as represented by the SAC). The genotypes from these simulated populations were used to simulate three phenotypes, each with a different association signal. Therefore the aim of this study was to determine if the LAAA model will be able to robustly capture the true causal variants regardless of the source of the association or the complexity of the admixture compared to other frequently used GWAS models. We also compare other models commonly used in GWAS and we investigate the efficacy of each model when presented with true versus inferred ancestry.

## Material and methods

### Simulating genotypes

msprime [12] was used to jointly simulate the demographic history (Fig 1) of the Nama, South African Coloured (SAC), Han Chinese in Beijing, China (CHB), British From England and Scotland (GBR), Gujarati Indians in Houston, Texas, USA (GIH), Gumuz, Luhya in Webuye, Kenya (LWK) and Mende in Sierra Leone (MSL). The model describing the demographic history was defined in a Demes Specification (https://github.com/popsim-consortium/demes-python) that was then converted to a msprime demographic model using the demes python library. Demes Specifications are YAML files that contain information about populations, their properties (like effective population size changes) and the relationship of a population to other populations.

The demographic history of the Nama, an indigenous population of southern Africa [13], is complex and extends back to the emergence of modern humans. Furthermore, the Nama also integrated and received ancestral contributions from the indigenous Khoe-San group from southern Africa. The Khoe-San are reported to have the most divergent lineages compared to other living populations [14–18] and it is believed that they have largely remained isolated until ~2000 years ago [13, 14, 19]. The Nama is reported to have experienced 5–25% gene flow from eastern African caprid and cattle pastoralists ~2000 years ago [20]. These events were finally followed by recent admixture with European colonists ~250 years ago. The demographic model for the Nama (Fig 1) was based on the above mentioned events and a conference poster by Ragsdale et al. [21] where they inferred detailed parameterized demographic

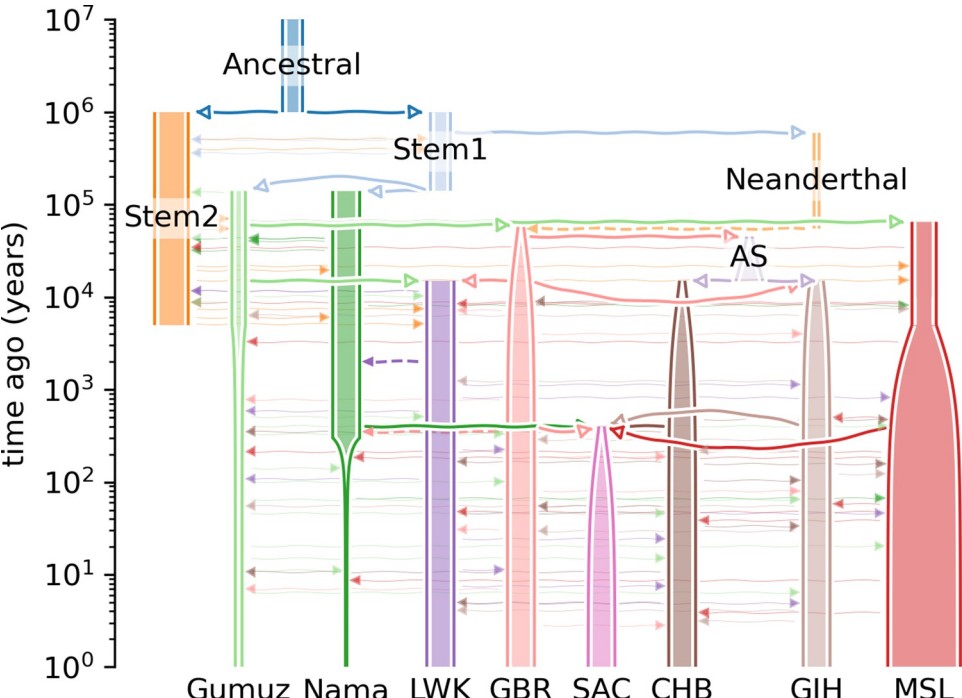

**Fig 1. The demographic histories of the simulated populations (Nama and SAC).** Thick solid lines with open arrowheads indicate an ancestor -> descendant relation, dashed lines indicate an admixture pulse and faint solid lines with closed arrowheads indicate continuous migration. The two admixture events for the Nama is indicated with the dashed lines with arrowheads (purple and light pink). The one admixture event for the SAC is indicated by the four solid lines with open arrowheads (green, dark pink, light pink and light brown).

# Simulations

## Simulating genotypes with msprime

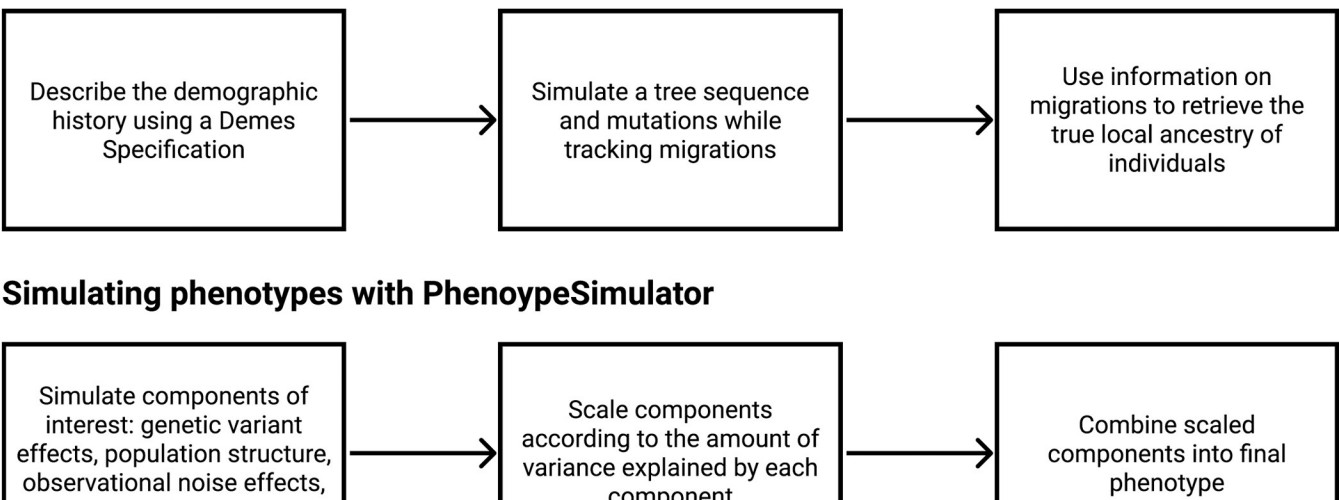

**Fig 2. Overview of the methods used to simulate genotypes using the software msprime and phenotypes using the software PhenotypeSimulator.**

models for five present day populations (GBR, Gumuz, LWK, MSL and Nama) using joint allele frequency and LD statistics. The Khoe-San also contribute a significant ancestral component to a South African population that represents a highly admixed group from multiple ancestral populations (SAC) [22, 23]. Many non-African and African populations moved into southern Africa over the last ~600 years and integrated with the Khoe-San [22]. Uren *et al.* [13] found that the SAC has Khoe-San (32–43%), Bantu-speaking African (20–36%), European (21–28%) and East and South-East Asian (9–11%) ancestral contributions. The demographic model for the SAC (Fig 1) was based on the above ancestral proportions with a single admixture event, for simplicity, between the ancestral populations ~400 years ago.

The demographic models were used to simulate DNA sequence evolution of 5000 Nama and 5000 SAC individuals as tree sequences (Fig 2). We assumed a generation time of 25 years, a mutation rate of 1.29e-08 and a recombination rate as specified by the Combined HapMap II recombination map. The CHB, GBR, GIH, LWK and MSL (1000 individuals of each) were simulated to act as reference populations for the Nama and the SAC. The Nama acted as a reference population for any Khoe-San ancestral contribution in the Nama and the SAC. Our aim with this simulation was not to create models that accurately represent the demographic history of the Nama or the SAC, but to simulate realistic genotype data for a three-way and five-way admixed population as represented by the Nama and the SAC. The tree sequence was then used to create VCFs using tskit [24]. The number of SNPs in the resultant VCFs was reduced to ~400 000 SNPs to reduce model runtime by randomly selecting variants. We also wrote a custom script that works its way up the tree sequence to find the ancestry at a given locus at a given time using the migration history of that locus, thereby enabling us to recover the true ancestry of an individual at a given locus.

## Simulating phenotypes

PhenotypeSimulator [25] was used for phenotype simulation, because it allows very fine control over individual effects. It entails a three-step process: (1) simulating phenotype components of interest, (2) scaling the components according to the amount of variance explained and (3) combining the phenotype components for each individual. The phenotype components are further divided into genetic effects and noise effects. Genetic effects are made up of genetic variant effects, population structure, local ancestry and the interaction between variant effects and local ancestry effects. Noise effects are made up of observational noise effects and confounding variable such as sex and age.

Three kinds of phenotypes were simulated for the Nama (Table 1) and the SAC (Table 2)—phenotypes with an allelic only (AO) effect, phenotypes with an ancestry plus allelic (APA) effect and phenotypes with an APA and local ancestry adjusted allelic (LAAA) effect. Three replications, each with a different set of 10 causal SNPs of each kind of phenotype were simulated randomly and a replication was simulated with each ancestral population (three for the Nama and five for the SAC) as the source of the local ancestry effect. Therefore, in total 27 phenotypes were simulated for the Nama and 45 phenotypes we simulated for the SAC.

For all phenotypes, genetic effects were assumed to account for 60% of the total variance. When genetic variant effects were present, they were assumed to account for 20% of the total genetic effect or 12% of the total effect, when local ancestry effects are present, they were assumed to account for 10% of the total genetic effect or 6% of the total effect and when the interactions between variant effects and local ancestry effects were present, they were assumed to account for 6.67% of the total genetic effect or 4% of the total effect. After these effects were accounted for, the proportion of the genetic effect that remains was attributed to population structure and kinship. Observational noise was assumed to be 40% of the total noise effect or 16% of the total effect and the non-genetic covariates, sex and age, was assumed to account for the remaining 60% of the noise effect or 24% of the total effect. These proportions are completely arbitrary and only the genetic variant effects are based on literature—complex diseases, such as type 2 diabetes and systemic lupus erythematosus, have been associated with 4–40 loci with a total heritability explained by variant effects of 1.5–50% [26].

## Ancestry inference

The simulated Nama and SAC datasets were merged with their respective appropriate source populations using PLINK v2.0 (https://www.cog-genomics.org/plink/2.0/) [27] in order to infer both global and local ancestry. For the Nama, the GBR (n = 1000), LWK (n = 1000) and

**Table 1. Breakdown of the effect sizes that make up the phenotypes simulated for the Nama.**

| Phenotype | Ancestry of interest | Genetic effects | | | | Noise effects | |
| | | Variant | Local ancestry | Interaction | Population structure | Observational | Covariates |
|---|---|---|---|---|---|---|---|
| AO | Nama | 0.12 | 0.00 | 0.00 | 0.48 | 0.16 | 0.24 |
| AO | GBR | 0.12 | 0.00 | 0.00 | 0.48 | 0.16 | 0.24 |
| AO | EP | 0.12 | 0.00 | 0.00 | 0.48 | 0.16 | 0.24 |
| APA | Nama | 0.12 | 0.06 | 0.00 | 0.42 | 0.16 | 0.24 |
| APA | GBR | 0.12 | 0.06 | 0.00 | 0.42 | 0.16 | 0.24 |
| APA | EP | 0.12 | 0.06 | 0.00 | 0.42 | 0.16 | 0.24 |
| LAAA | Nama | 0.12 | 0.06 | 0.04 | 0.38 | 0.16 | 0.24 |
| LAAA | GBR | 0.12 | 0.06 | 0.04 | 0.38 | 0.16 | 0.24 |
| LAAA | EP | 0.12 | 0.06 | 0.04 | 0.38 | 0.16 | 0.24 |

**Table 2. Breakdown of the effect sizes that make up the phenotypes simulated for the SAC.**

| Phenotype | Ancestry of interest | Genetic effects | | | | Noise effects | |
|---|---|---|---|---|---|---|---|
| | | Variant | Local ancestry | Interaction | Population structure | Observational | Covariates |
| AO | Nama | 0.12 | 0.00 | 0.00 | 0.48 | 0.16 | 0.24 |
| AO | GBR | 0.12 | 0.00 | 0.00 | 0.48 | 0.16 | 0.24 |
| AO | MSL | 0.12 | 0.00 | 0.00 | 0.48 | 0.16 | 0.24 |
| AO | CHB | 0.00 | 0.00 | 0.00 | 0.60 | 0.16 | 0.24 |
| AO | GIH | 0.00 | 0.00 | 0.00 | 0.60 | 0.16 | 0.24 |
| APA | Nama | 0.12 | 0.06 | 0.00 | 0.42 | 0.16 | 0.24 |
| APA | GBR | 0.12 | 0.06 | 0.00 | 0.42 | 0.16 | 0.24 |
| APA | MSL | 0.12 | 0.06 | 0.00 | 0.42 | 0.16 | 0.24 |
| APA | CHB | 0.12 | 0.06 | 0.00 | 0.42 | 0.16 | 0.24 |
| APA | GIH | 0.12 | 0.06 | 0.00 | 0.42 | 0.16 | 0.24 |
| LAAA | Nama | 0.12 | 0.06 | 0.04 | 0.38 | 0.16 | 0.24 |
| LAAA | GBR | 0.12 | 0.06 | 0.04 | 0.38 | 0.16 | 0.24 |
| LAAA | MSL | 0.12 | 0.06 | 0.04 | 0.38 | 0.16 | 0.24 |
| LAAA | CHB | 0.12 | 0.06 | 0.04 | 0.38 | 0.16 | 0.24 |
| LAAA | GIH | 0.12 | 0.06 | 0.04 | 0.38 | 0.16 | 0.24 |

Nama (n = 1000) were included to represent their contributing ancestral source populations. For the SAC, the GBR (n = 1000), MSL (n = 1000), Nama (n = 1000), GIH (n = 1000) and CHB (n = 1000) were included to represent their contributing ancestral source populations. After merging of admixed and source ancestral populations, all minor alleles with a frequency < 0.05 were excluded from analysis. The final dataset after quality control and data filtering consisted of 387 959 autosomal variants and 5000 Nama and 5000 SAC, in addition to 3000 ancestral individuals for the Nama and 5000 ancestral individuals for the SAC. The software RFMix was used to infer global and local ancestry for both the Nama and SAC datasets [28]. RFMix is 30X faster than other local ancestry inference software and is accurate in multi-way admixture scenarios [28, 29]. Default parameters were used, except for the number of generations since admixture, which was set to 13 for the Nama and 16 for the SAC.

## GWAS models

The following five regression models were tested simultaneously for each simulated phenotype for both the Nama and SAC:

1. <u>The standard model</u>—only allelic effects are considered and no global or local ancestry effects are modeled.

$$E(Y) = \alpha_o + \alpha_1 E_1 + \alpha_2 E_2 + \beta X_x$$

**$E(Y)$** is the continuous outcome ($Y$). **$E_1$** and **$E_2$** represent the covariates (age and gender) and $\alpha_1$ and $\alpha_2$ is the corresponding marginal effect. $X_x$ represents the number of reference alleles at the locus under investigation and $\beta$ is the corresponding marginal effect.

2. <u>The global ancestry (GA) model</u>—frequently used in GWAS to account for the possible confounding effect of population structure in the study population.

$$E(Y) = \alpha_o + \alpha_1 E_1 + \alpha_2 E_2 + \alpha_p P + \beta X_x$$

**$P$** represents the estimated global ancestry and $\alpha_p$ is the corresponding marginal effect.

3. <u>The local ancestry (LA) model</u>—used in admixture mapping studies to localise potential associations to ancestral blocks. The model is sensitive to frequency disparities across ancestral populations.

$$E(Y) = \alpha_o + \alpha_1 E_1 + \alpha_2 E_2 + \alpha_p P + y X^i$$

$X^i$ represents the number of ancestry alleles (separate model for each contributing ancestry) at the locus under investigation and $y$ is the corresponding marginal effect.

4. <u>The ancestry plus allelic (APA) model</u>—tests for both an allelic association and an association with an ancestry in the phenotype. Therefore jointly testing the associations modelled in model 1 + 2.

$$E(Y) = \alpha_o + \alpha_1 E_1 + \alpha_2 E_2 + \alpha_p P + \beta X_x + y X^i$$

5. <u>The local ancestry adjusted allelic (LAAA) model</u>—an extension of the APA model that also includes an interaction term between the allele present at a specific locus and the ancestry.

$$E(Y) = \alpha_o + \alpha_1 E_1 + \alpha_2 E_2 + \alpha_p P + \beta X_x + y X^i + \eta X^i_x$$

$X^i_x$ represents the number of ancestry-specific reference alleles at the locus under investigation and $\eta$ is the corresponding marginal effect.

When global ancestry is included in a given model, the smallest ancestry proportion was excluded as a covariate to avoid complete separation of the data for both the Nama (GBR) and SAC (CHB). Thus, two ancestral components (Nama and LWK) for the Nama and four ancestral components (Nama, MSL, GBR, GIH) for the SAC were included as covariates in association testing, together with gender and age. A total of 5000 Nama and 5000 SAC and 387 959 autosomal variants were included in statistical analysis. The lm() function in $R$ was used for linear regression association testing. The conventional significance threshold of $5 \times 10^{-8}$ for association testing was used. Dosage files were compiled at each locus for the allelic state (0, 1 or 2 copies of the major allele), for the ancestry (0, 1 or 2 copies of the ancestry of interest) and for the interaction between the allelic state and the ancestry (0, 1 or 2 copies of the major allele that is from the ancestry of interest).

## Results

### Local ancestry inference accuracy

We inferred the local ancestry for the simulated Nama and SAC populations, since the true local ancestry will not be available in a GWAS using real data. Therefore, we could also assess the impact of LAI accuracy on the GWAS models in different phenotypic scenarios. An overall 84.90% accuracy for the Nama (Table 3) and an overall 85.40% accuracy for the SAC was attained. For the ancestral components of the Nama we attained 76.07% for the LWK, 88.35% for the GBR and 84.90% for the Nama (Khoe-San). For the ancestral components of the SAC

**Table 3. The ancestral proportions and the accuracy of the inferred local ancestry for the Nama.**

|  | LWK | GBR | Nama | Total |
|---|---|---|---|---|
| Ancestral proportion | 0.100 | 0.150 | 0.750 | 1.000 |
| LAI accuracy | 0.761 | 0.884 | 0.853 | 0.849 |

**Table 4. The ancestral proportions and the accuracy of the inferred local ancestry for the SAC.**

|  | CHB | GBR | MSL | Nama | GIH | Total |
|---|---|---|---|---|---|---|
| Ancestral proportion | 0.100 | 0.175 | 0.250 | 0.350 | 0.125 | 1.000 |
| LAI accuracy | 0.831 | 0.892 | 0.846 | 0.847 | 0.854 | 0.854 |

(Table 4) we attained 83.13% for the CHB, 89.24% for the GBR, 84.56% for the MSL, 84.73% for the Nama and 85.38% for the GIH.

## Comparison of GWAS models applied to simulated GWAS data for the Nama and the SAC

We compared the true positive (the number of times when the source of association could be correctly identified) and the false positive hits (the number of times when the source of the association was incorrectly identified as a true association) from each GWAS model applied to all of the simulated phenotypes to get an indication of model success rate in the simulated Nama and SAC populations. Three iterations of each phenotype were simulated with different sets of causal SNPs, the GWAS models were run for each iteration and the number of true positive and false positive hits (excluding sites in LD with causal SNPs) were averaged to produce the results in Figs 3–6. Across all phenotypes in both populations, whether true (Figs 4 and 6) or inferred (Figs 3 and 5) ancestry was used, the LAAA model achieved the highest or tied for the highest true positive hit count. However, the AO phenotypes in the Nama (Figs 3 and 4) sometimes identified more true positive hits using the standard model, but this only for the AO phenotype. As expected, the standard model was tied with the LAAA model for the highest hit count in phenotypes with only an allelic effect (Figs 5 and 6) in the SAC. In phenotypes that have an ancestry effect in the Nama, either the APA model or the standard model achieved the second highest hit count after the LAAA model (Figs 3 and 4). However, in the SAC the APA model identified more true positive hits in all phenotypes (Figs 5 and 6). The LA model did not identify any true positive hits in many cases and produced a substantial number of false positive hits when there is an ancestry effect present (Figs 3–6). The GA model identified more true positives than the LA model, but fewer than the standard model. The standard model produced the lowest number of false positive hits regardless of the phenotype it was applied to, but when there is an ancestry effect present, the standard model never identified more than 6/10 causal variants whereas models such as LAAA sometimes identified 9/10.

All models produced a low number of false positive hits for the AO phenotypes relative to the other phenotypes. Overall, models with a local ancestry component (APA, LA, LAAA) produced substantially more false positive hits than models that only account for allelic effects and global ancestry effects (Figs 3–6). There are more false positive hits across phenotypes and models when inferred ancestry was used for the Nama and the SAC instead of the true ancestry (Figs 3–6). For the SAC there are slightly more false positive hits for phenotypes with the CHB and GBR as the source of the ancestral effect when using the true ancestry (Fig 6) compared to the inferred ancestry (Fig 5). In every phenotype, except the AO phenotypes, the LWK had the lowest number of false positive hits in the Nama (Figs 3 and 4) and the CHB had the lowest number of false positive hits in the SAC when the inferred ancestry was used (Fig 6). This pattern holds true for the other inferred ancestries (except the GIH) in the SAC phenotypes (Fig 5). In the Nama, the use of inferred local ancestry in the APA, LA and GA models (Fig 3) also produced more false positive hits in the AO phenotypes than using the true ancestry in these models. Using inferred ancestry over the true ancestry noticeably decreased the number of true positive hits detected by the APA, LA and the GA, whereas the LAAA model is less affected by inaccurate inferred local ancestry.

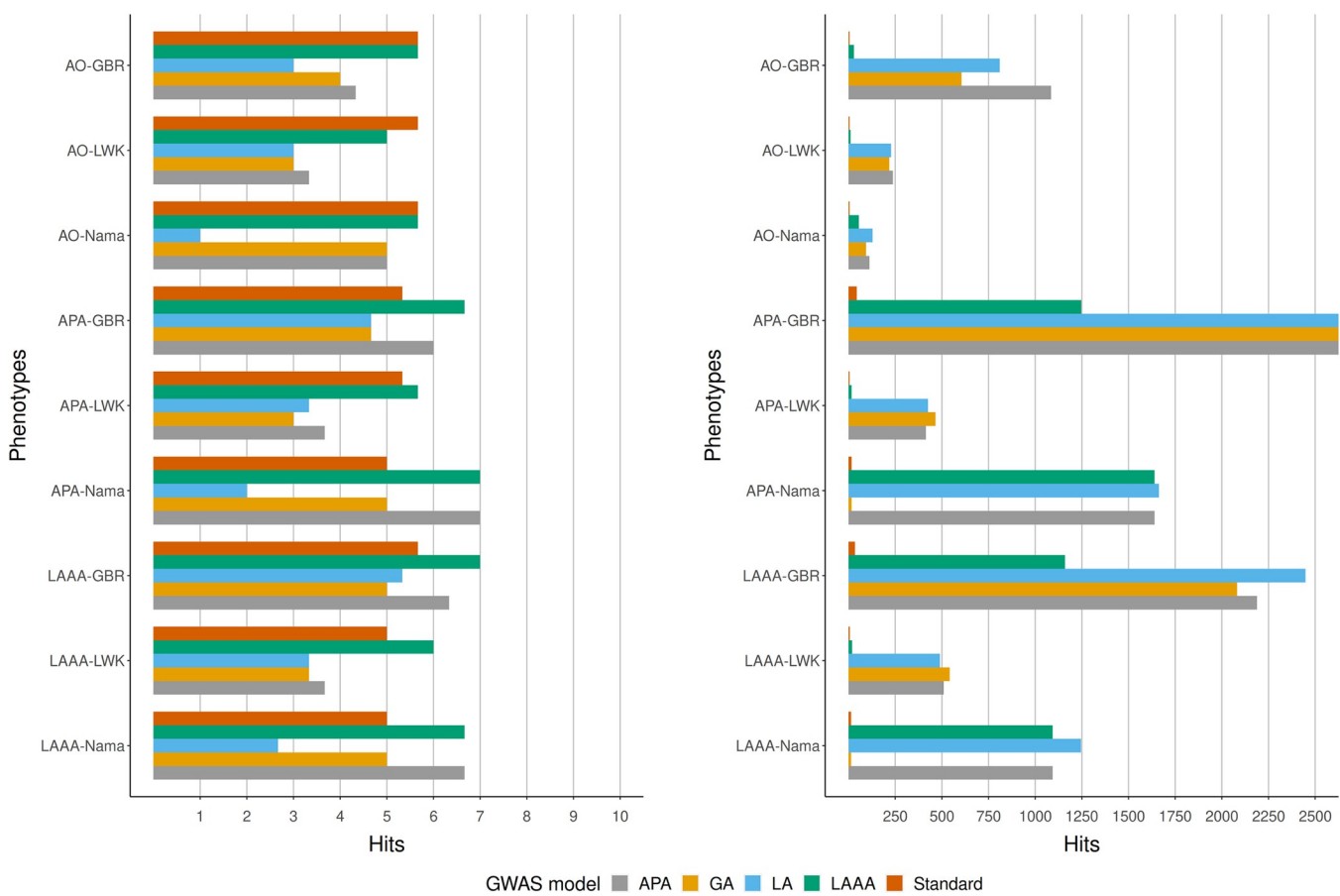

**Fig 3. Comparison of true positive hits (left) and false positive hits (right) for the Nama with inferred local ancestry used in GWAS models.** The average hits for three runs with different causal SNPs are shown. The simulated phenotypes are denoted as "phenotype-ancestral source of association", e.g., LAAA-LWK means the LAAA phenotype with the LWK ancestral component as the ancestral source of association. The various GWAS models used are indicated in different colours. Grey represents the APA model, yellow represents the GA model, blue represents the LA model, green represents the LAAA model and the orange represents the Standard model.

## Discussion

Generally, researchers adjust for the possible confounding effect of population structure by considering global ancestry proportions or top principal components in statistical models. Conversely, researchers would conduct admixture mapping, which involves inferring ancestry at each genomic loci along the genome of an admixed individual (local ancestry inference), to increase the power to detect variants with an ancestry effect [30, 31]. This is sufficient in simple admixture scenarios, such as two-way admixed populations (African-Americans) and three-way admixed populations (Latinos/Hispanics) [1, 32, 33]. Populations from southern Africa, however, can be complex multi-way admixed populations, with up to five contributing ancestries [34, 35]. Furthermore, the underlying genetic architecture is mostly unknown and can be complex due to various historical events in southern Africa [4]. Thus, conventional admixture mapping strategies might result in either increased false positive hits or false negative hits for complex multi-way admixed populations. Duan et al. [8] first described LAAA analysis as a robust method for discovering association signals, while producing few false positive hits, by comparing the LAAA model to other well-known models used for GWAS in admixed populations [8]. Their simulation study, however, was limited to a two-way admixed population.

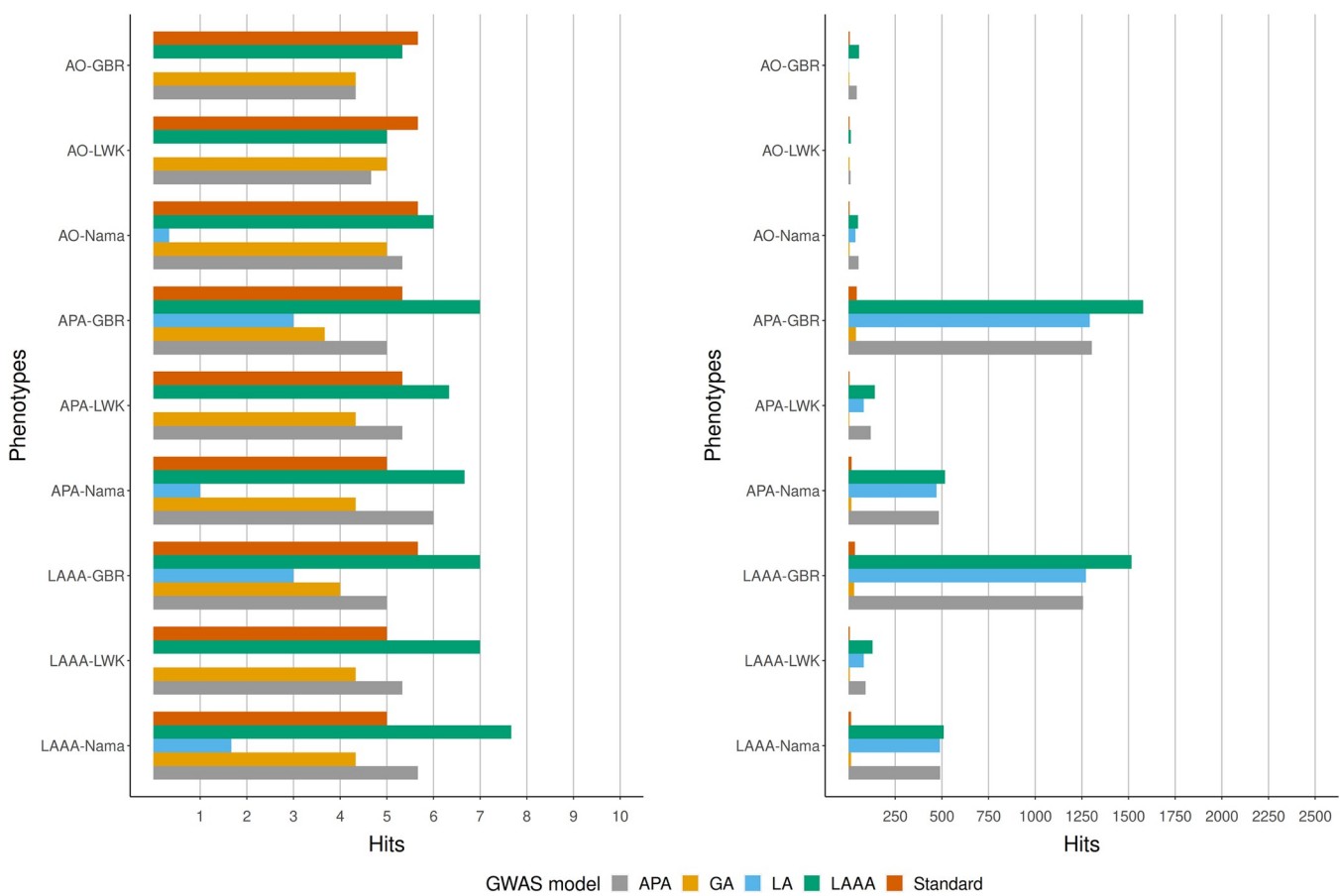

**Fig 4. Comparison of true positive hits (left) and false positive hits (right) for the Nama with true local ancestry used in GWAS models.** The average hits for three runs with different causal SNPs are shown. The simulated phenotypes are denoted as "phenotype-ancestral source of association", e.g.,LAAA-LWK means the LAAA phenotype with the LWK ancestral component as the ancestral source of association. The various GWAS models used are indicated in different colours. Grey represents the APA model, yellow represents the GA model, blue represents the LA model, green represents the LAAA model and the orange represents the Standard model.

Realising that their findings might not translate to other admixture scenarios, we simulated a three- and five-way admixed population to compare the GA, LA, APA and LAAA models that they tested and added a conventional GWAS model that does not model any local ancestry effects (the standard model). Our main objective is was to discern if the LAAA model robustly captures the true causal variants in complex multi-way admixed populations without prior knowledge of admixture dynamics compared to other traditional GWAS models.

The GA model is the most widely used model to control for population structure in GWAS, because it has been shown to improve GWAS results when the phenotype exhibits a differential allelic effect across ancestral populations (scenario AO). Therefore, adjusting for only global ancestry proportions or the top principle components should be sufficient to control for the confounding effect of population structure. In phenotypes where there is no ancestry effect, we expected that the GA model would improve upon the standard model, however, this was not the case.

The LA model identified the least amount of true positive causal variants and sometimes none at all. This makes sense, since only local ancestry is considered in statistical analysis and the possible allelic effect is not accounted for. The LA model is more successful in phenotypes where the ancestral component that causes the ancestry effect is small, which is expected, since

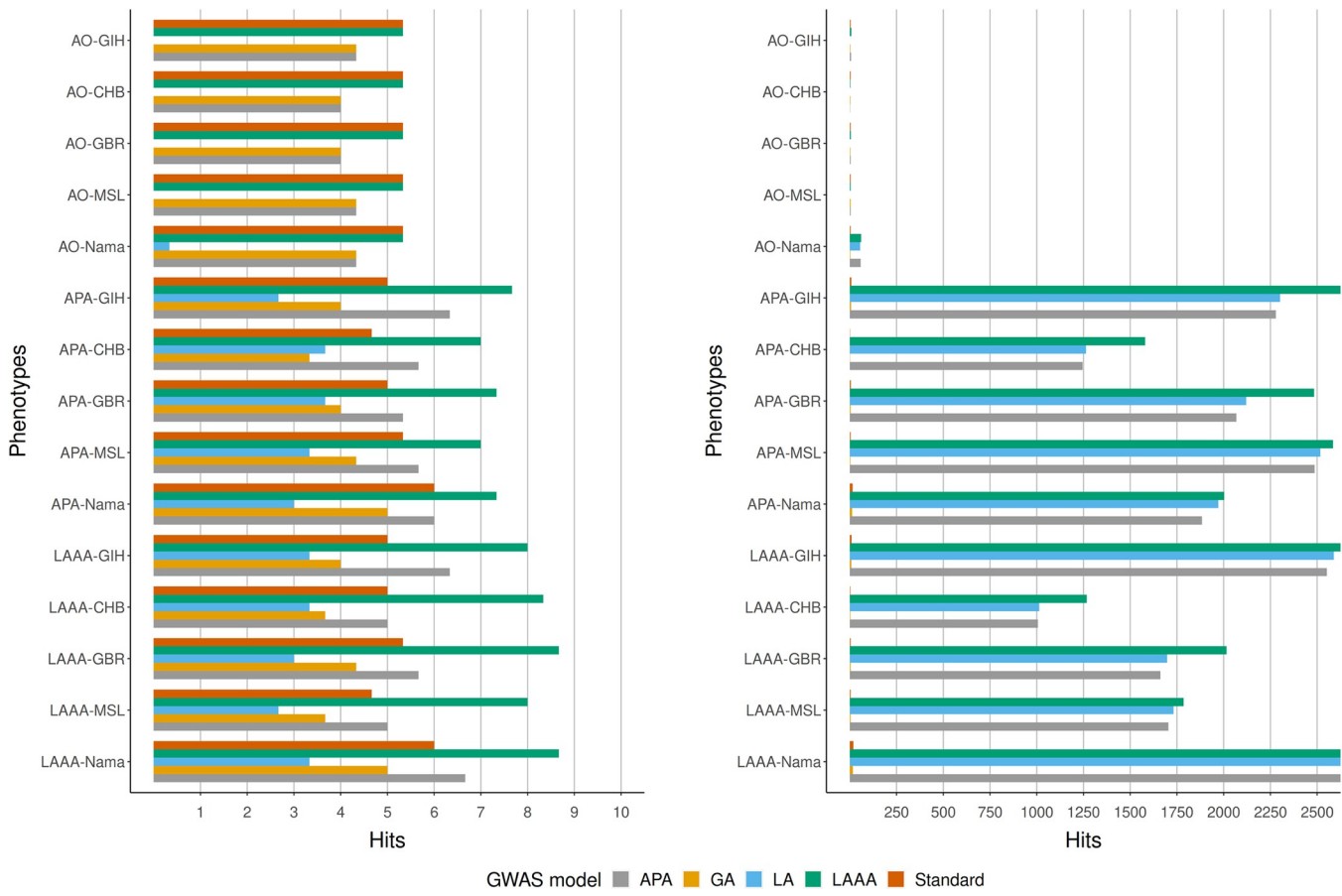

**Fig 5. Comparison of true positive hits (left) and false positive hits (right) for the SAC with inferred local ancestry used in GWAS models.** The average hits for three runs with different causal SNPs are shown. The simulated phenotypes are denoted as "phenotype-ancestral source of association", e.g., LAAA-CHB means the LAAA phenotype with the CHB ancestral component as the ancestral source of association. The various GWAS models used are indicated in different colours. Grey represents the APA model, yellow represents the GA model, blue represents the LA model, green represents the LAAA model and the orange represents the Standard model.

the effect is concentrated in fewer SNPs. We only observed this effect in the five-way admixed population and not in the three-way admixed population (Figs 4 and 6). This effect could be due to chance, since causal SNPs were drawn randomly from the dataset without any regard for the local ancestry of the SNP. Our simulations would have to be adjusted to ensure that all of the causal SNPs are found in or outside of the contributing ancestry's ancestral LD blocks and many more iterations would have to be done to test this. Any success in identifying true causal variants with this method is negated by the large number of false positive hits that are produced. The number of false positive hits produced by this model are generally organised into admixture peaks where all the SNPs that share an ancestral LD block with the causal variant will have a strong association with the phenotype. This approach only indicates the ancestral LD block associated with the disease trait and, therefore, requires more information to identify which variant is responsible for the association perceived in the admixture peak. Furthermore, when the simulated local ancestry effects are sufficiently large, over inflation of test statistics could occur.

The APA model is expected to perform better than the first two models when there is both an allelic and ancestry effect, because both of these effects are modelled at each genomic locus. When testing genetic markers that are proxies for a disease causal loci, the differential LD

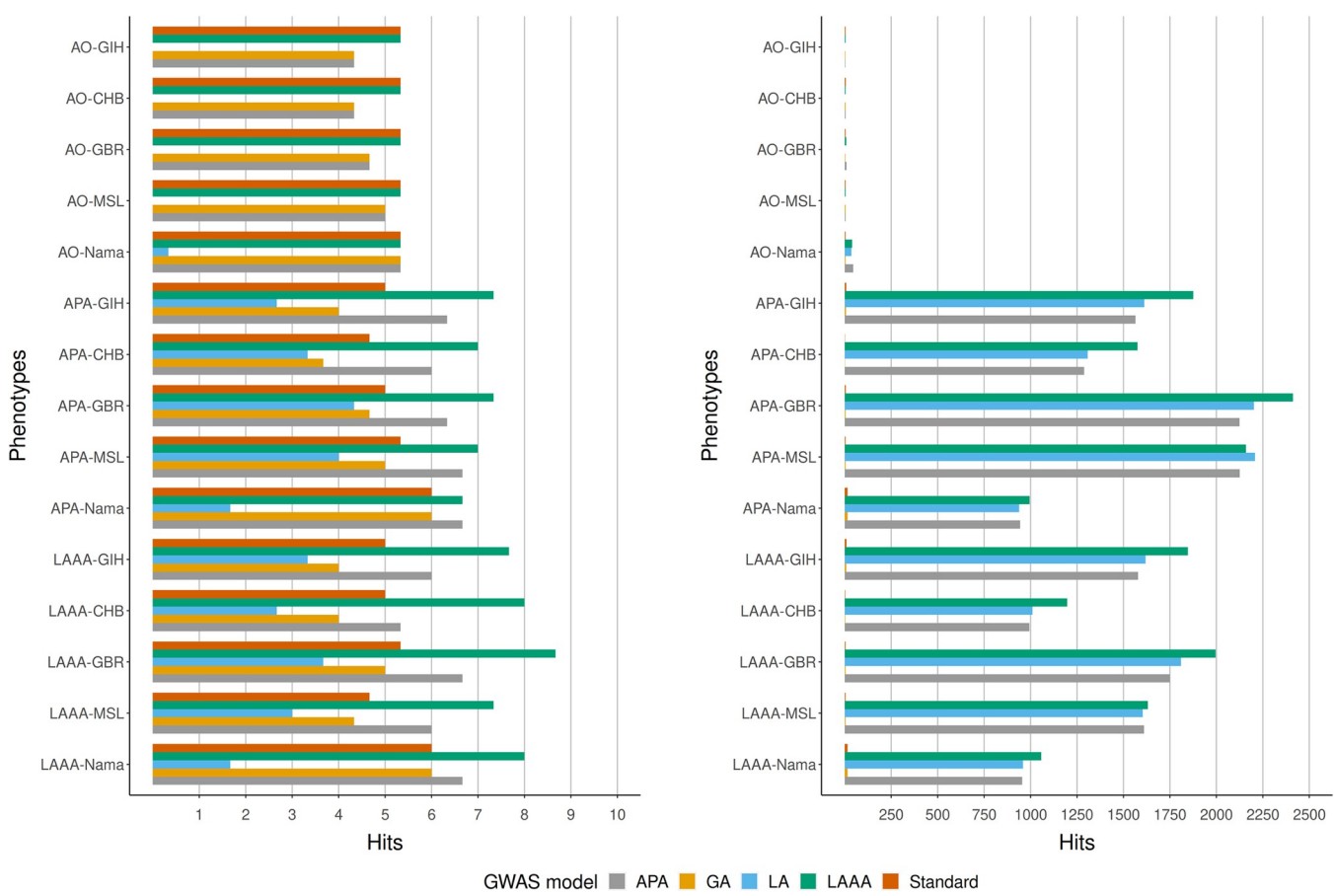

**Fig 6. Comparison of true positive hits (left) and false positive hits (right) for the SAC with true ancestry used in GWAS models.** The average hits for three runs with different causal SNPs are shown. The simulated phenotypes are denoted as "phenotype-ancestral source of association", e.g.,LAAA-CHB means the LAAA phenotype with the CHB ancestral components as the ancestral source of association. The various GWAS models used are indicated in different colours. Grey represents the APA model, yellow represents the GA model, blue represents the LA model, green represents the LAAA model and the orange represents the Standard model.

within admixed populations can result in heterogeneity of effect estimates by local ancestry [36]. Therefore, additional adjustments beyond global ancestry are required to increase power when the admixture-induced LD is in the opposite direction as the LD in the ancestral population [8, 36]. This agrees with what we observe from our results where the GA model identified fewer true positive hits in comparison to the APA and LAAA models when the simulated phenotypes displayed a local ancestry effect. The APA model is also more readily affected by LAI accuracy than the LAAA model. Hence, the APA model could be more prone to false negative hits and false positive hits. The APA model might also be insufficient for certain admixture scenarios, such as scenarios where admixture induced LD blocks mask the allelic effect of the causal SNP when the tagging SNP is located in a different ancestral LD block to the causal SNP.

The LAAA model outperforms the other statistical models by either achieving the most true positive hits or achieving the second highest hit count. Most importantly, the LAAA is particularly robust in the presence of genetic effect heterogeneity. Interestingly, Duan et al. [8] and Liu et al. [36] both recommend to first identify admixture peaks and then to use the interaction term as a secondary follow-up test to elucidate the source of the association obtained in the admixture peak. However, our results demonstrate the possible true causal variants that

might be missed when not including the interaction term from the beginning in complex multi-way admixed populations. Importantly, the LAAA model was robust against inaccurate local ancestry inference (0.849 for the Nama and 0.854 for the SAC) with only minor changes in the number of true positive hits between GWAS performed with inferred ancestry versus true ancestry.

We assumed independence of ancestry-specific effects across chromosomes. Therefore, the effect of other ancestral LD blocks (besides the ancestral LD blocks in which the causal SNPs are found) on a given phenotype was not simulated. This is important to note when considering our results in terms of real data, since this omission can have an impact on elucidating the true causal SNPs. Future studies should consider including all five ancestries in a linear mixed model to account for cases where traits are affected by the interaction between multiple ancestral LD blocks and their interaction with the causal SNPs. Another unexpected observation in the results, is the large number of false positive hits for both the APA and LAAA models compared to the other three models (Standard, GA and LA). This could be due to the simulated local ancestry effect size being too large, causing associations of whole ancestral blocks with the trait of interest. This study was limited by only conducting three replications of the simulated genotypes and phenotypes. Ideally, we would perform a few thousand replications to allow us to calculate the power of each model to identify causal SNPs. However, we were limited by the computational resources required for such an experiment. Alternatively, we could have reduced the number of SNPs used in the analysis and we could have simulated a single population instead of two. The impact of changes of the mutation rate and recombination rate on the statistical models ability to capture true variants could also be tested in the future. The simplification of the SAC as one admixture event could have implications for the admixture-induced LD blocks, since colonisation and admixture that occurred in the Cape was complex, at different time points and still ongoing. Hence, the simulations should be expanded to multiple admixture events at different time points in the future.

In conclusion, investigating admixed individuals can be advantageous, due to diverse patterns of LD across continental populations. Nevertheless, the differential allele frequencies and LD patterns that exist between the ancestries should be incorporated correctly for true causal variants to be identified. The findings of this study show that jointly testing an allele effect, ancestry effect, and allele effect heterogeneity across ancestry (LAAA) in a regression model guards against missing genuine associations from any of the three sources. However, it is evident from the large number of false positive hits detected by some models, that the ancestry effect size we assumed across our simulated phenotypes was too large. Therefore, we were unable to fully assess the suitability of the LAAA model in a complex multi-way admixture scenario. Considering the extent to which the various models we tested differed in their results and considering that the source of a given association is unknown, we recommend that researchers use multiple GWAS models when analysing populations with complex ancestry. Specifically when the population under study exhibits extensive genetic diversity and complicated LD patterns, which are often observed for southern African populations.

## Acknowledgments

We would like to acknowledge the Centre for High Performance Computing (CHPC), South Africa, for providing computational resources.

## Author Contributions

**Conceptualization:** Yolandi Swart, Gerald van Eeden.

**Formal analysis:** Yolandi Swart, Gerald van Eeden.

**Investigation:** Yolandi Swart, Gerald van Eeden.

**Methodology:** Yolandi Swart, Gerald van Eeden.

**Supervision:** Caitlin Uren.

**Visualization:** Yolandi Swart, Gerald van Eeden.

**Writing – original draft:** Yolandi Swart, Gerald van Eeden.

**Writing – review & editing:** Caitlin Uren, Gian van der Spuy, Gerard Tromp, Marlo Möller.

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
