## [Decision Letter · Decision Letter 0]

30 May 2022

PONE-D-22-03783GWAS in the Southern African contextPLOS ONE

Dear Dr. Swart,

Thank you for submitting your manuscript to PLOS ONE. After careful consideration, we feel that it has merit but does not fully meet PLOS ONE’s publication criteria as it currently stands. Therefore, we invite you to submit a revised version of the manuscript that addresses the points raised during the review process.

 Please submit your revised manuscript by Jul 14 2022 11:59PM. If you will need more time than this to complete your revisions, please reply to this message or contact the journal office at plosone@plos.org. Please include the following items when submitting your revised manuscript:A rebuttal letter that responds to each point raised by the academic editor and reviewer(s). You should upload this letter as a separate file labeled 'Response to Reviewers'.A marked-up copy of your manuscript that highlights changes made to the original version. You should upload this as a separate file labeled 'Revised Manuscript with Track Changes'.An unmarked version of your revised paper without tracked changes. You should upload this as a separate file labeled 'Manuscript'.If applicable, we recommend that you deposit your laboratory protocols in protocols.io to enhance the reproducibility of your results. Protocols.io assigns your protocol its own identifier (DOI) so that it can be cited independently in the future. For instructions see: https://journals.plos.org/plosone/s/submission-guidelines#loc-laboratory-protocols. Additionally, PLOS ONE offers an option for publishing peer-reviewed Lab Protocol articles, which describe protocols hosted on protocols.io. Read more information on sharing protocols at https://plos.org/protocols?utm_medium=editorial-email&utm_source=authorletters&utm_campaign=protocols.

We look forward to receiving your revised manuscript.

Kind regards,

Badri Padhukasahasram

Academic Editor

PLOS ONE

**Journal requirements:**

“This research was partially funded by the South African

government through the South African Medical Research

Council and the National Research Foundation. YS was supported by a Stellenbosch

University Postgraduate Bursary.”

“This research was partially funded by the South African

government through the South African Medical Research

Council and the National Research Foundation. YS was supported by a Stellenbosch

University Postgraduate Bursary.”

6. We note that you have indicated that data from this study are available upon request. PLOS only allows data to be available upon request if there are legal or ethical restrictions on sharing data publicly. For more information on unacceptable data access restrictions, please see http://journals.plos.org/plosone/s/data-availability#loc-unacceptable-data-access-restrictions.

**Additional Editor Comments:**

Dear Dr. Swart,

The reviews for your manuscript GWAS in the Southern African context are now available.

Based on comments from 3 expert reviewers, I recommend that the manuscript be accepted provided authors make minor revisions outlined by all reviewers. In particular, changes to the text to make it clearer, explanation of definitions of true and false positives, better figure and figure legends can improve the manuscript. In terms of major comments, authors can consider a simplified 3-way and 5-way admixture scenarios to be simulated and tested to avoid additional confounding effects on the GWAS methods tested. Lastly, reviewers also request that the model and scripts used be made readily available to the readers.

I look forward to a revised manuscript that addresses all these comments and concerns.

Sincerely,

Dr. Badri Padhukasahasram

Reviewers' comments:

Reviewer's Responses to Questions

**Comments to the Author**

1. Is the manuscript technically sound, and do the data support the conclusions?

Reviewer #1: Partly

Reviewer #2: Yes

Reviewer #3: Yes

2. Has the statistical analysis been performed appropriately and rigorously? 

Reviewer #1: Yes

Reviewer #2: Yes

Reviewer #3: Yes

3. Have the authors made all data underlying the findings in their manuscript fully available?

Reviewer #1: Yes

Reviewer #2: Yes

Reviewer #3: Yes

4. Is the manuscript presented in an intelligible fashion and written in standard English?

Reviewer #1: Yes

Reviewer #2: Yes

Reviewer #3: Yes

5. Review Comments to the Author

Reviewer #1: The authors set out to validate a recently published approach to use admixture LD to improve GWAS in more complex admixture scenarios (3-way and 5-way). The authors use realistic effect sizes for genetic effects maximizing the applicability of their results to real GWAS situations. However, because of the complex demographic history of the source populations for the admixed communities used as models in this study, it is hard to disentangle effects from varying levels of a particular ancestry and those from a difference in diversity from a single ancestry. The paper has the potential to be a broadly useful resource to the genetic epidemiology community, and could be improved if a simplified 3-way and 5-way admixture scenario were simulated and tested to avoid additional confounding effects on the GWAS methods tested. However, that may be beyond the scope of what the authors hope to present here and I would not require it for accepting this manuscript.

I do find several issues with the paper that I think need to be addressed. While GWAS models are well specified (lines 225-253), the conditions of the simulated datasets (both phenotypes and genotypes) are not. It is difficult to evaluate the paper's claims without being able to evaluate how the data were produced.

1. I would suggest the authors include links to the model files used and the custom script used for ancestry assignment (line 161) from the simulated data should be linked to or provided as supplementary data.

2. It is unclear if the authors simulated all the admixture events (arrows) represented in figure 1.

3. It is unclear to me how the genotypes/phenotypes are being linked for analysis? How were the 10 causal SNPs chosen?

4. The three types of phenotypes simulated need to be explained in more detail. The discussion (lines 388-390) seems to suggest that a phenotype caused by a local ancestry-specific SNP was not simulated and tested with the various methods. Would distributing causal SNPs randomly across ancestries not result in biased results of all methods tested that are looking for an ancestry-specific effect? Could this explain the high number of false positives seen in the APA and LAAA models compared to the Standard and GA?

5. The relationship alluded to in Lines 316-318 between the amount of ancestry and false positives should be more formalized, for example simulating datasets with 5-95% of a certain ancestry and plotting the change in false positives. This would be broadly informative beyond the study populations used here.

6. On Line 271, how was LAI accuracy inferred? Were inaccurate segments identified and removed or were they misassigned?

7. The results of this study showing that in many instances the Standard model performs as well or nearly as well as the APA or LAAA, while providing orders of magnitude fewer false positives. Identifying 2-3 fewer true hits but avoiding ~1000 false positives that would make interpreting GWAS results extremely difficult, so to support the use of only a Standard Model. This is quite counter-intuitive and requires more discussion.

Reviewer #2: The author is not consistent with abbreviation and full name (under abstract GWAS is not describes) then later LAAA is used as abbreviation and as a full term.

The aim and the rational of this study its not clearly stated.

The Author is using a public data but it was not mentioned clearly under method section that, this is a public data and therefore there is no need for ethical clearance.

I think there is a need for a paragraph of sample/data description.

Figures have poor quality (they are blur).

All Figure legends have incomplete description of what is happening in a figure.

I think sub title will make it easier to follow.

Reviewer #3: 1 This is an interesting and self-contained paper that will be very helpful to people analysing complex populations. The authors simulated 3-way and 5-way admixture populations that are broadly representative of two groups in South Africa and then with different phenotypes try to understand which models correctly detect SNPs associated with the phenotypes. There are several different modelling approaches and as seen in the paper how you do this has a significant timpact

2 The paper is written clearly and overall the methods are sound.

3 I have one major criticism

4 There is no definition of true positive and false positive -- see detailed comments below. Without knowing this, I am not sure I agree with the conclusions (or at least I can't independently say I do). If their definition of a false positive would include a SNP that is _not_ a SNP in the "input" of the simulation as being causal but is for example 20 base pairs away and has been detected because of LD effects then I think that needs considerable motivation and explanation to justify calling that a false positive. Anyway -- please be clear and define this.

5. The number of SNPs used is only 400k -- which is small -- does this impact the result? If we were using a dense array and/or imputed data would this help or make things more complicated

6. The limitations expressed on page 15 are significant There are different results for 3-way and 5-way admixture (and as the authors point out this differs from results by others for 2-way admixture). But is that a function of the n in n-way or is it more complex than that? I suppose I want a more definitive answer, which is not possible with the evidence givern.

7. As a point of principle I think the code that was developed should be made available via GitHub or the like.

8. Bibliography -- a very annyoing feature of the paper is that references are numbered in the text but there is no number in the bibliography!!!!

9 Other issues:

l97-98: There is no reference for the claim that two-way admixed populations "generally" have simple demographic histories and originate from a single pulse admixture event. I suppose this depends what is meant by "simple" and "pulse", but I think either give some reference that supports this or weaken the claim.

l117 : "From" -> from

Comma after "Khoe-San" on l128 should deleted -- the subject of the sentence is complex and hence the temptation but the comma is wrong and it would be better to simplify the txt.

l122 : See point above about availability -- shouldn't these specifications be made available publicly. Just from a selfish point of view you are more likely to get citations if you do and people use them.

l 126-130 : I think the characterisation of Khoe-San as a group culturally related needs care as this is contested esp by members of the communities. This is not my area, but the term Khoe-san language is a term that is more convenient for people outside those communities rather than accurate (perhaps like speaking about the Danish-Japanese language gruop?) and I don't think you can say that pastoralism and hunter-gatherer lifestyles are close culturally. As a relatively recent phenomenon as a result of colonialism since the the late 18th C this may have become true in recent historical times but to extrapolate from one place may be risk over-generalising.

l142: wouldn't South Asian be better than East Asian if you are using GIH as the proxy.

l143: This modelling of SAC is as the authors recognise a simplification -- white settlement in the Cape only started ~370 years ago with ongoing admixture -- and probably at peak from 300-150 years ago. This is not intended as a criticism and I don't think this result weakens the results but I think the authors should address the consequences of this simplification.

l167 Simulating phenotypes: A little more detail would be useful here -- I'd like to understand how this takes LD and both global and local ancestry into account. I didn't understand how the SNPs were chosen -- do you just choose one lead SNP in a region or do you simulate SNPs that though not causal are in LD with the "causal" SNP. This para ia bit opaque. Is different LD in different popualtions considered.

l178 -- use dash not hyphen (l 201, 227, 238 and many other palces too)

l 288-307 -- I don't think true positives etc should be hyphenated

l299 -- semi-colon not comma at end

l378 -- principle -> principal

l383 -- amount -> number

l395 -- I am intrigued by this. When we do a GWAS we often get hits caused by SNPs being in LD wth the causal (or at least a lead SNP). We don't see these as false hits though because we understand that they tag the causal SNP. Of course this depends on how bug the admixture block is -- I wonder whether your framework of true positives/false positives is correct, and I think this needs clarificaition at least. If you define a false positive as a SNP that is found in the simulated data output then I think it is too strong -- I think this related to my comment on line 167 above -- see my general comment.

l436 -- no comma -- the subject is complex but it just makes it worse to put a comma in. Similar issue on line 456

l448 I don't think only more complex admixture scenarios. Also just understanding the variability you get that may depend on changes in admixture ratios, LD-structure, population difference and so on.

l 462. This sentence needs a main verb.

6. PLOS authors have the option to publish the peer review history of their article (what does this mean?). If published, this will include your full peer review and any attached files.

Reviewer #1: No

Reviewer #2: No

Reviewer #3: No

---

## [Author Response · Author response to Decision Letter 0]

5 Aug 2022

Thank you for your review and comments on our research manuscript. Please find our point-by-point response to the Reviewers comments below. 

Corrections to the conform to the journal requirements have been made as suggested. This include:

- Figures were regenerated and submitted to (PACE) digital diagnostic tool to ensure that figures meet PLOS requirements. 

- Figure legends were updated. 

- Sentence reconstruction where more clarity was required. 

- Bibliography was updated to showcase numbers as required

Amendment to ‘Funding Statement’: Please including the following as our funding statement. There is no specific grant number attached to this, but the phrasing has been approved the South African Medical Research Council.

This research was funded (partially or fully) by the South African government through the South African Medical Research Council and the National Research Foundation. GvE was supported by the DST-NRF Innovation Doctoral Scholarship. YS was supported by a Stellenbosch University Postgraduate Bursary. There was no additional external funding received for this study. The funders had no role in study design, data collection, and analysis, decision to publish, or preparation of the manuscript.

Amendment tot Data availability statement: 

All genetic data used in this manuscript was simulated or retrieved from public repositories (http://www.internationalgenome.org). All scripts used to generate and conduct data analysis are deposited and publicly available on Github: https://github.com/YolandiSwart/GWAS_SA

The following was done to improve the overall quality of the paper as recommended by reviewers: 

- Explanation of true and false positives

o We added an explanation in lines 282-285, “We compared the true positives (the number of times the source of association could be correctly identified) and the false positives (the number of times the source of the association was incorrectly identified as a true association) from each GWAS model applied to all of the simulated phenotypes to get an indication of model success rate in the simulated Nama and SAC populations.”

- Consider a simplified 3-way and 5-way admixture scenarios to be simulated and tested to avoid additional confounding effects on the GWAS methods tested. 

o Thank you for this comment. We agree with the reviewer that this would be interesting and we indeed considered including more simplified 2-way, 3-way and 5-way admixed populations. However, just with the factors we considered we already simulated 24 distinct phenotypes with three different causal sets for each and tested 5 GWAS models on each of those iterations. Thus, 360 GWAS was run in total. This was computationally intensive and we were restricted with computational resources. Thus, we focused on the most realistic scenario one would encounter if you would conduct GWAS analysis on South African populations. 

- The model and scripts made readily available to the readers 

o All scripts used in the manuscript were made publicly available via a Github link. (https://github.com/YolandiSwart/GWAS_SA)

Responses to each reviewers comments: 

Reviewer #1: 

1. I would suggest the authors include links to the model files used and the custom script used for ancestry assignment (line 161) from the simulated data should be linked to or provided as supplementary data.

Thank you for this comment. All scripts are now publicly available via a github link. (https://github.com/YolandiSwart/GWAS_SA)

2. It is unclear if the authors simulated all the admixture events (arrows) represented in figure 1.

Thank you for this comment. No, not all admixture events were simulated that are represented in figure 1. To clear up any confusion we added two additional sentences in lines 133-136. 

“The two admixture events for the Nama are indicated with the dashed lines with arrowheads (purple and light pink). Four admixture events were simulated at one time point for the SAC. These are indicated by the four solid lines with open arrowheads (green, dark pink, light pink and light brown).”

3. It is unclear to me how the genotypes/phenotypes are being linked for analysis? How were the 10 causal SNPs chosen?

Thank you for this comment. The genotypes are linked with the phenotypes when conducting regression model analysis for each simulated phenotype to see if there is a significant association between the simulated phenotype and causal variant (represented as the continuous outcome (Y)). 

The choice of 10 causal SNPs are explained in lines 169-172. The 10 causal SNPs are randomly simulated, since one would not know before conducting a GWAS where the causal SNP would be located. 

“Three replications, each with a different set of 10 causal SNPs of each kind of phenotype were simulated randomly and a replication was simulated with each ancestral population (three for the Nama and five for the SAC) as the source of the local ancestry effect.” 

This is also confirmed in the discussion in lines 411-413, “This effect could be due to chance, since causal SNPs were drawn randomly from the dataset without any regard for the local ancestry of the SNP.”

4. The three types of phenotypes simulated need to be explained in more detail. 

Thank you for this comment. We have added a thorough explanation of the simulated three phenotypes in lines 167-173. Two tables (Table 1 for Nama and Table2 for the SAC) are included to showcase the breakdown of the effect sizes of the simulated phenotypes. The scripts on Github will also clarify the confusion. 

“Three phenotypes were simulated for the Nama (Table 1) and the SAC (Table 2) - phenotypes with an allelic only (AO) effect, phenotypes with an ancestry plus allelic (APA) effect and phenotypes with an APA and local ancestry adjusted allelic (LAAA) effect. Three replications, each with a different set of 10 causal SNPs of each kind of phenotype were simulated randomly and a replication was simulated with each ancestral population (three for the Nama and five for the SAC) as the source of the local ancestry effect. Therefore, in total 27 phenotypes were simulated for the Nama and 45 phenotypes were simulated for the SAC.”

The discussion (lines 388-390) seems to suggest that a phenotype caused by a local ancestry-specific SNP was not simulated and tested with the various methods. Would distributing causal SNPs randomly across ancestries not result in biased results of all methods tested that are looking for an ancestry-specific effect? Could this explain the high number of false positives seen in the APA and LAAA models compared to the Standard and GA?

Thank you for this comment. The goal of this manuscript was to test if the LAAA robustly capture the true causal variants regardless of the source of the association or the complexity of the admixture compared to other frequently used GWAS models. We intended to distribute causal SNPs across ancestries, because this is the most realistic scenario that we witness in admixed South African populations. Hence, the models used should be robust in any admixture scenario. We also included an explanation for the high number of false-positives in the discussion in lines 449-451, 

“Another unexpected observation in the results, is the large number of false positive hits for both the APA and LAAA models compared to the other three models (Standard, GA and LA). This could be due to the simulated local ancestry effect size being too large, causing associations of whole ancestral blocks with the trait of interest.”

5. The relationship alluded to in Lines 316-318 between the amount of ancestry and false positives should be more formalized, for example simulating datasets with 5-95% of a certain ancestry and plotting the change in false positives. This would be broadly informative beyond the study populations used here.

Thank you for this comment. We agree that it would be an interesting analysis to do, but the level of heterogeneity in the simulated data already encompasses a wide range of admixture which is more representative of modern populations. This will become even more applicable in the future. 

6. On Line 271, how was LAI accuracy inferred? Were inaccurate segments identified and removed or were they misassigned?

Thank you for this comment. To test the LAI accuracy of a given run, the inferred ancestry was compared to the true ancestry per locus per individual. Segments were not removed as we wanted to assess the effects of misassigned tracks on the models (Figure 3-6). This is evident in the scripts added to the Github page (https://github.com/YolandiSwart/GWAS_SA/blob/main/Ancestry_inference/lai_accuracy_per_anc.R). 

7. The results of this study showing that in many instances the Standard model performs as well or nearly as well as the APA or LAAA, while providing orders of magnitude fewer false positives. Identifying 2-3 fewer true hits but avoiding ~1000 false positives that would make interpreting GWAS results extremely difficult, so to support the use of only a Standard Model. This is quite counter-intuitive and requires more discussion.

Thank you for this comment. In the discussion in lines 451-453 we explain that the amount of false positives seen in the APA or LAAA and not in the AO models, is most probably due to the ancestry effect size that we simulated and not due to the model itself. 

Reviewer #2: The author is not consistent with abbreviation and full name (under abstract GWAS is not describes) then later LAAA is used as abbreviation and as a full term.

Thank for this comment. We apologise for this inconsistency. Abbreviations were corrected throughout the manuscript. 

The aim and the rational of this study it’s not clearly stated.

Thank for this comment. The aim and rationale are now stated in line 92. 

“Therefore the aim of this study was to determine if the LAAA model will be able to robustly capture the true causal variants regardless of the source of the association or the complexity of the admixture. ”

The Author is using a public data but it was not mentioned clearly under method section that, this is a public data and therefore there is no need for ethical clearance.

Thank for this comment. All genetic data used in this manuscript was simulated or retrieved from public repositories (http://www.internationalgenome.org). All scripts used to generate and conduct data analysis are deposited and publicly available on Github: https://github.com/YolandiSwart/GWAS_SA

I think there is a need for a paragraph of sample/data description.

Thank for this comment. In lines 158-194 we explain the sample/data simulated for this study with two descriptive tables (Table 1 and Table 2), as well as included the scripts to generate this datasets on Github. https://github.com/YolandiSwart/GWAS_SA

Figures have poor quality (they are blur).

Thank for this comment. We re-uploaded figures to ensure good quality. 

All Figure legends have incomplete description of what is happening in a figure.

Thank for this comment. We agree and expanded the descriptions of all figures throughout the manuscript. 

Fig 1. The demographic histories of the simulated populations (Nama and SAC). Thick solid lines with open arrowheads indicate an ancestor -> descendant relation, dashed lines indicate an admixture pulse and faint solid lines with closed arrowheads indicate continuous migration. The two admixture events for the Nama are indicated by the dashed lines with arrowheads (purple and light pink). Four admixture events were simulated at one time point for the SAC. These are indicated by the four solid lines with open arrowheads (green, dark pink, light pink and light brown). 

Fig 2. Overview of the methods used to simulate genotypes using the software msprime and phenotypes using the software PhenotypeSimulator.

Fig 3. Comparison of true positives (left) and false positives (right) for the Nama with inferred local ancestry used in GWAS models. The average hits for three runs with different causal SNPs are shown. The simulated phenotypes are denoted as “phenotype-ancestral source of association”, e.g.,LAAA-LWK means the LAAA phenotype with the LWK ancestral component as the ancestral source of association. The various GWAS models used are indicated in different colours. Grey represents the APA model, yellow represents the GA model, blue represents the LA model, green represents the LAAA model and the orange represents the Standard model. 

Fig 4. Comparison of true positive hits (left) and false positive hits (right) for the Nama with true local ancestry used in GWAS models. The average hits for three runs with different causal SNPs are shown. The simulated phenotypes are denoted as “phenotype-ancestral source of association”, e.g.,LAAA-LWK means the LAAA phenotype with the LWK ancestral component as the ancestral source of association. The various GWAS models used are indicated in different colours. Grey represents the APA model, yellow represents the GA model, blue represents the LA model, green represents the LAAA model and the orange represents the Standard model. 

Fig 5. Comparison of true positive hits (left) and false positive hits (right) for the SAC with inferred local ancestry used in GWAS models. The average hits for three runs with different causal SNPs are shown. The simulated phenotypes are denoted as “phenotype-ancestral source of association”, e.g.,LAAA-CHB means the LAAA phenotype with the CHB ancestral component as the ancestral source of association. The various GWAS models used are indicated in different colours. Grey represents the APA model, yellow represents the GA model, blue represents the LA model, green represents the LAAA model and the orange represents the Standard model. 

Fig 6. Comparison of true positive hits (left) and false positive hits (right) for the SAC with inferred local ancestry used in GWAS models. The average hits for three runs with different causal SNPs are shown. The simulated phenotypes are denoted as “phenotype-ancestral source of association”, e.g.,LAAA-CHB means the LAAA phenotype with the CHB ancestral components as the ancestral source of association. The various GWAS models used are indicated in different colours. Grey represents the APA model, yellow represents the GA model, blue represents the LA model, green represents the LAAA model and the orange represents the Standard model. 

Reviewer #3: 1 This is an interesting and self-contained paper that will be very helpful to people analysing complex populations. The authors simulated 3-way and 5-way admixture populations that are broadly representative of two groups in South Africa and then with different phenotypes try to understand which models correctly detect SNPs associated with the phenotypes. There are several different modelling approaches and as seen in the paper how you do this has a significant timpact

2 The paper is written clearly and overall the methods are sound.

3 I have one major criticism

4 There is no definition of true positive and false positive -- see detailed comments below. Without knowing this, I am not sure I agree with the conclusions (or at least I can't independently say I do). If their definition of a false positive would include a SNP that is _not_ a SNP in the "input" of the simulation as being causal but is for example 20 base pairs away and has been detected because of LD effects then I think that needs considerable motivation and explanation to justify calling that a false positive. Anyway -- please be clear and define this.

Thank you for this comment. 

We added defined true and false positives in lines 282-285, “We compared the true positives (the number of times the source of association could be correctly identified) and the false positives (the number of times the source of the association was incorrectly identified as a true association) from each GWAS model applied to all of the simulated phenotypes to get an indication of model success rate in the simulated Nama and SAC populations.”

Indeed, we do not consider a SNP in LD with a true causative variant as a false positive, as pointed out in the discussion (line 399):, “Our simulations would have to be adjusted to ensure that all of the causal SNPs are found in or outside of the contributing ancestry’s ancestral LD blocks and many more iterations would have to be done to test this.”

This is also further discussed in lines 438-442, “We assumed independence of ancestry-specific effects across chromosomes. Therefore, the effect of other ancestral LD blocks (besides the ancestral LD blocks in which the causal SNPs are found) on a given phenotype was not simulated. This is important to note when considering our results in terms of real data, since this omission can have an impact on elucidating the true causal SNPs.” 

5. The number of SNPs used is only 400k -- which is small -- does this impact the result? If we were using a dense array and/or imputed data would this help or make things more complicated

Thank you for this comment. Yes, increasing the number of SNPs would indeed make things more complicated and required more computational resources. We have previously investigated imputation accuracy and showed we that imputation in southern African populations are feasible and produces quality data (Schurz et al. 2019, doi: 10.3389/fgene.2019.00034). Due to the complexity and computational expense of the work presented in this manuscript, we chose not to use imputed data to introduce more statistical bias. Increasing the number of SNPs in our simulations would not have improved the accuracy, as we knew where the causal SNPs were.

6. The limitations expressed on page 15 are significant There are different results for 3-way and 5-way admixture (and as the authors point out this differs from results by others for 2-way admixture). But is that a function of the n in n-way or is it more complex than that? I suppose I want a more definitive answer, which is not possible with the evidence givern.

Thank you for this comment. Each admixture scenario will be different due to historical events. Most African populations had integration from various continental ancestral populations and the exact and location of ancestry proportions are mostly unknown (genetic heterogeneity). Hence this is more complex than the n in n-way, since ancestry patterns may be different between admixed populations formed by the same source populations.

7. As a point of principle I think the code that was developed should be made available via GitHub or the like.

Thank you for this comment. We agree with this statement and all scripts are now publicly available via a github link. https://github.com/YolandiSwart/GWAS_SA

8. Bibliography -- a very annyoing feature of the paper is that references are numbered in the text but there is no number in the bibliography!!!!

Thank you for comment. We apologise and corrected the numbers in the bibliography. 

9 Other issues:

l97-98: There is no reference for the claim that two-way admixed populations "generally" have simple demographic histories and originate from a single pulse admixture event. I suppose this depends what is meant by "simple" and "pulse", but I think either give some reference that supports this or weaken the claim.

Thank you for this comment. We have removed this statement.

l117 : "From" -> from – 

This was corrected 

Comma after "Khoe-San" on l128 should deleted -- the subject of the sentence is complex and hence the temptation but the comma is wrong and it would be better to simplify the txt.

Thank you for this comment. This sentence was restructured in lines 132-133 for easier reading.

 “Uren et al. (13) found that the SAC has Khoe-San (32-43%), Bantu-speaking African (20-36%) , European (21-28%) and East and South-East Asian (9-11%) ancestral contributions.”

l122 : See point above about availability -- shouldn't these specifications be made available publicly. Just from a selfish point of view you are more likely to get citations if you do and people use them.

Thank you for this comment. All scripts are now publicly available via a github link. https://github.com/YolandiSwart/GWAS_SA

l 126-130 : I think the characterisation of Khoe-San as a group culturally related needs care as this is contested esp by members of the communities. This is not my area, but the term Khoe-san language is a term that is more convenient for people outside those communities rather than accurate (perhaps like speaking about the Danish-Japanese language gruop?) and I don't think you can say that pastoralism and hunter-gatherer lifestyles are close culturally. As a relatively recent phenomenon as a result of colonialism since the the late 18th C this may have become true in recent historical times but to extrapolate from one place may be risk over-generalising.

Thank you for this comment. The term Khoe-San encompasses both Khoekhoe (pastoralist) and San (hunter-gatherer) populations. Khoe-San refers to the two population groups collectively, rather than referring to a language group. During the course of our research on these communities, we have seen that this term has largely been accepted by these communities but each have their own identity. We however agree that our previous statement was confusing and have rephrased the sentence for clarity in lines 110-114. 

“The demographic history of the Nama, an indigenous Khoe-San population of southern Africa (13), is complex and extends back to the emergence of modern humans. Similar to the SAC, the Nama also received ancestral contributions from a number of non-African populations. “ 

l142: wouldn't South Asian be better than East Asian if you are using GIH as the proxy.

Thank you for this comment. We used GIH as proxy for South Asian in line 141. 

l143: This modelling of SAC is as the authors recognise a simplification -- white settlement in the Cape only started ~370 years ago with ongoing admixture -- and probably at peak from 300-150 years ago. This is not intended as a criticism and I don't think this result weakens the results but I think the authors should address the consequences of this simplification.

Thank you for this comment. We agree with this statement and added an additional discussion in lines 455-459. 

“The simplification of the SAC as multiple admixture events simultaneously could have implications for the admixture-induced LD blocks, since colonisation and admixture that occurred in the Cape was complex, at different time points and still ongoing. Hence, the simulations should be expanded to multiple admixture events at different time points in the future.” 

l167 Simulating phenotypes: A little more detail would be useful here -- I'd like to understand how this takes LD and both global and local ancestry into account. I didn't understand how the SNPs were chosen -- do you just choose one lead SNP in a region or do you simulate SNPs that though not causal are in LD with the "causal" SNP. This para ia bit opaque. Is different LD in different popualtions considered.

Thank you for this comment. LD was not simulated in the different populations. Line 288 specifies this: “positives and false positives (excluding sites in LD with causal SNPs) were averaged to”. 

The choice of 10 causal SNPs are explained in lines 167-173. The 10 causal SNPs are randomly simulated, since one would not know before conducting a GWAS where the causal SNP would be located. 

We have added a thorough explanation of the simulated three phenotypes in lines 167-173. Two tables (Table 1 for Nama and Table2 for the SAC) are included to showcase the breakdown of the effect sizes of the simulated phenotypes. The scripts on Github will also clarify the confusion. 

“Three phenotypes were simulated for the Nama (Table 1) and the SAC (Table 2) - phenotypes with an allelic only (AO) effect, phenotypes with an ancestry plus allelic (APA) effect and phenotypes with an APA and local ancestry adjusted allelic (LAAA) effect. Three replications, each with a different set of 10 causal SNPs of each kind of phenotype were simulated randomly and a replication was simulated with each ancestral population (three for the Nama and five for the SAC) as the source of the local ancestry effect. Therefore, in total 27 phenotypes were simulated for the Nama and 45 phenotypes were simulated for the SAC.”

Both global and local ancestry are only inferred after simulation of genotypes. Although the ancestral populations contributing to the admixed population is simulated as stipulated in lines 138-151. 

“The demographic model for the Nama (Fig 1) was based on the above mentioned events and a conference poster by Ragsdale et al. [21] where they inferred detailed parameterized demographic models for five present day populations (GBR, Gumuz, LWK, MSL and Nama) using joint allele frequency and LD statistics. Khoe-San groups also contributed a significant ancestral component to many South African populations, including a highly admixed group from multiple ancestral populations (SAC) [22][23]. Many non-African and African populations moved into southern Africa over the last ~600 years and integrated with Khoe-San groups [22]. Uren et al. [13] found that the SAC has Khoe-San (32-43%), Bantu-speaking African (20-36%) , European (21-28%) and East and South Asian (9-11%) ancestral contributions. The demographic model for the SAC (Fig 1) was based on the above ancestral proportions with a single admixture event, for simplicity, between the ancestral populations ~400 years ago.

We made the following corrections to the manuscript and apologise for these errors:

l178 -- use dash not hyphen (l 201, 227, 238 and many other palces too)

All dashes were changed to hyphens.

l 288-307 -- I don't think true positives etc should be hyphenated 

We removed the hyphen in all instances

l299 -- semi-colon not comma at end 

We have corrected this.

l378 -- principle -> principal

Corrected

l383 -- amount -> number 

Corrected

l436 -- no comma -- the subject is complex but it just makes it worse to put a comma in. Similar issue on line 456 

Corrected

l 462. This sentence needs a main verb. 

A main verb was added

l395 -- I am intrigued by this. When we do a GWAS we often get hits caused by SNPs being in LD wth the causal (or at least a lead SNP). We don't see these as false hits though because we understand that they tag the causal SNP. Of course this depends on how bug the admixture block is -- I wonder whether your framework of true positives/false positives is correct, and I think this needs clarificaition at least. If you define a false positive as a SNP that is found in the simulated data output then I think it is too strong -- I think this related to my comment on line 167 above -- see my general comment.

Thank you for this comment, yes we do not see these as false positives. We stated that we excluded sites in LD with causal SNPs in line 283. “positive and false positive hits (excluding sites in LD with causal SNPs) were averaged to”. 

We also expanded on the definition of false positive hits and true positive hits in our study context in lines 282-284. 

“We compared the true positive (the number of times when the source of association could be correctly identified) and the false positive hits (the number of times when the source of the association was incorrectly identified as a true association) from each GWAS model applied”

This is also further discussed in lines 438-442, “We assumed independence of ancestry-specific effects across chromosomes. Therefore, the effect of other ancestral LD blocks (besides the ancestral LD blocks in which the causal SNPs are found) on a given phenotype was not simulated. This is important to note when considering our results in terms of real data, since this omission can have an impact on elucidating the true causal SNPs.” 

l448 I don't think only more complex admixture scenarios. Also just understanding the variability you get that may depend on changes in admixture ratios, LD-structure, population difference and so on.

Thank you for this comment. We agree with this statement. We removed this statement in lines 455-458. 

We thank all reviewers for this opportunity to revise and improve our manuscript.

---

## [Editor Report · Decision Letter 1]

8 Aug 2022

GWAS in the Southern African context

PONE-D-22-03783R1

Dear Dr. Swart,

We’re pleased to inform you that your manuscript has been judged scientifically suitable for publication and will be formally accepted for publication once it meets all outstanding technical requirements.

Kind regards,

Badri Padhukasahasram

Academic Editor

PLOS ONE
---

## [Editor Report · Acceptance letter]

6 Sep 2022

PONE-D-22-03783R1 

GWAS in the Southern African context 

Dear Dr. Swart:

I'm pleased to inform you that your manuscript has been deemed suitable for publication in PLOS ONE. Congratulations! Your manuscript is now with our production department. 

Kind regards, 

on behalf of

Dr. Badri Padhukasahasram 

%CORR_ED_EDITOR_ROLE%

PLOS ONE